# Replication Experiments and Microstructural Evolution of the Ancient Co-Fusion Steelmaking Process

**Shangxiao Qiao ***  **and Wei Qian**

Institute of Cultural Heritage and History of Science and Technology, University of Science and Technology Beijing, Beijing 100083, China; qianwei@ustb.edu.cn
* Correspondence: qiaoshangxiao115@163.com; Tel.: +86-152-0146-3866

**Abstract:** The study of co-fusion was one of the essential topics in the history of metallurgy in China. Simulation experiments were an essential concept in the study of the co-fusion steelmaking process. This paper mainly studied the simulation experiments of co-fusion from two aspects: the replication of co-fusion swords by three different methods and the micro-analysis of the co-fusion samples. The experimental results indicated that several co-fusion swords could be made by different processes, but the carbon content and surface hardness were quite different. During repeated forge welding, the microstructure of the samples transformed from laminated to homogenized; finally, a steel with a uniform carbon content was obtained. It was challenging to determine the characteristics of co-fusion from the homogenized samples. The results prompt a rethinking of the microstructural characteristics of ancient co-fusion artifacts.

**Keywords:** history of metallurgy; co-fusion steelmaking; ancient China; microstructure evolution; simulation experiments; cast iron; wrought iron

---

## 1. Introduction

In the ancient world, there were significant differences in iron and steel technology systems between the East and the West. The invention and extension of cast iron and cast-iron steelmaking technology was the core divergence before the 14th century CE. The metallurgical system based on cast iron laid a continuity of civilization and a solid foundation for national integration and unity in ancient China [1].

There were two main processes of steelmaking from cast iron: decarburization and co-fusion. The term "co-fusion" was first proposed by Joseph Needham in 1955; it corresponded to smelting carbon steel from the combination of cast iron and wrought iron [2]. Unlike modern wrought iron, the term "wrought iron" here meant that the carbon content of iron was too low to be steel, but it did not contain any information about the smelting technology. Co-fusion was mainly used to produce "edge steel" which was essential for making weapons and tools in ancient times, and it had excellent hardness and toughness. The research on the microstructures of swords is one of the hotpoint issues at present, and scholars have analyzed them all over the world [3,4]. Different cultures had significant differences in the processes of smelting swords' steel in ancient times such as the "Wootz" process in India and Sri-Lanka [5,6] and the "Bulat" process in Central Asia [7–9] to smelt crucible steel, the "Zuku-oshi" process to smelt cast iron, and the "Kera-oshi" process to smelt crude steel in Japan [10]. There is no report on the production of steel by smelting cast iron with wrought iron, and co-fusion is represents this process. However, the technical details of the co-fusion process have been lost, and the microstructural characteristics of the co-fusion samples have not evident until now.

Scholars have researched co-fusion from the perspective of simulation experiments and microstructural analysis to replicate this ancient technology. The first research was carried out by Needham and Whitaker [2] in 1955, and a steel ingot containing 0.8% carbon content was obtained. Miao [11] tested several archaeological samples and did simulation experiments based on the analysis results. Finally, they obtained the experimental results with a microstructure similar to archaeological samples. Wagner defined co-fusion as two different processes based on smelting temperature, and the eutectic temperature was the determining factor of co-fusion [12,13]. Furthermore, it had a crucial effect on the microstructure characteristic. Ke et al. [14], Chen et al. [15], Jia et al. [16], and Liu et al. [17] analyzed the suspected archaeological co-fusion samples and discussed the determination criteria.

Identification of a particular steelmaking process associated with each iron artifact would be vital in the reconstruction of ancient iron technology. Thus, it was imperative to understand the microstructural changes in the co-fusion steelmaking process clearly. Pevious studies have shown that research on co-fusion steelmaking was still insufficient. It was not able to summarize the process of co-fusion steelmaking and the microstructural characteristics of co-fusion samples. The imperfection in understanding the co-fusion process led to a lack of understanding of the microstructures.

In this paper, to discuss the process of co-fusion steelmaking and the microstructural evolution of samples, a series of experiments were designed and carried out. The experiments could be divided into two parts: the replication of co-fusion swords by three different processes and the analysis of the microstructures of co-fusion samples.

## 2. Materials and Methods

### 2.1. Samples' Preparation

According to the results on the composition and microstructure of ancient cast iron and wrought iron, mainly referred to by Han [1] and Miao [11], the hypoeutectic white cast iron with low Si and low S was mainly selected in this experiment. The chemical composition of the materials is shown in Table 1.

**Table 1.** Chemical composition of the material for experiments (wt.%).

| Material | C | Si | Mn | P | S |
|----------|------|--------|-------|-------|-------|
| Cast iron | 3.75 | 0.06 | 0.015 | 0.11 | 0.10 |
| Wrought iron | 0.02 | ≤0.005 | ≤0.01 | 0.006 | 0.010 |

The cast iron was processed into a round piece with a diameter of 40 mm and a thickness of 3 mm. The wrought iron had the following shapes: a round piece with a diameter of 40 mm and a thickness of 5 mm; and an ingot with a length of 100 mm, a width of 40 mm, and a height of 5 mm. The prepared samples were cleaned then polished.

### 2.2. Experimental Procedure

Experiments of smelting and forge welding to replicate co-fusion swords were carried out with the help of Hu, an experienced swordsmith, in a swords workshop in Longquan County, Zhejiang Province in China. Based on the technological records in ancient Chinese literature, three different co-fusion processes were designed, then smelted, and forged welded into several swords. The furnace and forging temperature were measured at all times during the experiment. We used a platinum–rhodium thermocouple in the back center position of the furnace to gauge the furnace temperature, and a portable infrared thermometer Raytek 3i2ML3U (measuring range from 200 °C to 1800 °C, Everett, WA, USA) was used to detect the surface temperature of the steel ingot.

Samples were taken in each smelting and forge welding process to study the composition, surface hardness, and microstructural changes of the co-fusion samples during the whole process. A combustion-infrared absorption spectrum CS-2600 (NCS, Beijing, China) was mainly used for

macroscopic carbon content measurements. The surface hardness was measured by a TH-300 Rockwell hardness tester (Shidai, Beijing, China). Eleven points were measured for each sample, and the average value of the last 10 points was calculated. Microstructure observation and micro-analysis were conducted on samples collected in the swords workshop. The samples were prepared following standard metallographic procedures and etched using 3 vol.% nitric acid in balanced methanol. The microstructures were captured with a Leica DM4000M optical microscope (OM, Leica, Weztlar, Germany). The figures presented in this article to illustrate microstructures are optical micrographs unless otherwise stated.

## 3. Results

### 3.1. Replicating Co-Fusion Swords with Several Processes

#### 3.1.1. Co-Fusion Directly

The first process was direct co-fusion; it was one of the most straightforward processes. This process was to produce steel with a uniform carbon content by controlling the furnace's temperature and atmosphere. The upper limit of temperature in the furnace is controlled at 1300 °C by using a platinum–rhodium thermocouple detection. This was due to the fact that charcoal was used as fuel for forging in ancient China, and the upper-temperature limit was not high. The forging temperature ranged from 900 °C to 600 °C. After forging, shaping, and quenching, the length of this sword was 34.5 cm, and the weight was 280.45 g. The process flow, size, and pattern on the surface of co-fusion directly were displayed in Figure 1.

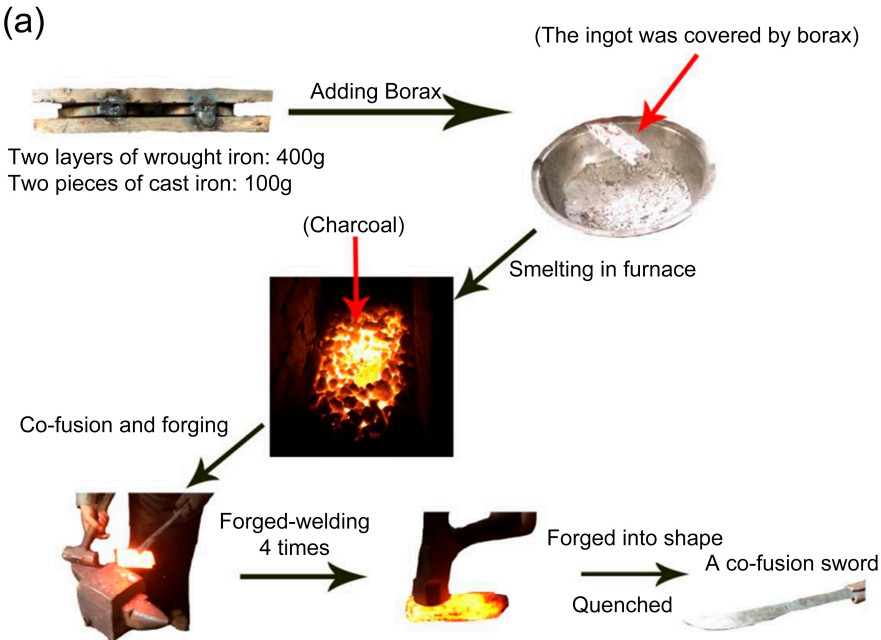

**Figure 1.** *Cont.*

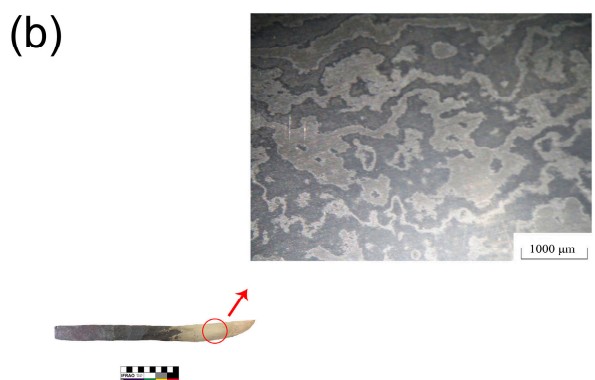

**Figure 1.** Direct co-fusion to make swords: (**a**) technological process; (**b**) the sword and pattern on the surface.

The first process was a relatively primitive process. From the positive aspect, the design of this process was simple. On the other hand, there were several deficiencies in this process:

1. It was tough to operate. Because of the uncertainty of temperature, atmosphere, and other factors in the furnace, the blacksmith must make a comprehensive judgment on the situation of the co-fusion steel ingot in the furnace, which provides a higher requirement for his experience;

2. There was a massive carbon loss in cast iron. Without a container, it was not easy to control the state of cast iron in this process. In order to ensure the efficiency of homogenization diffusion, the furnace temperature was often higher, which will cause the rapid melting and flow loss of cast iron, resulting in the reduction of carbon content in the swords;

3. The oxidation degree was challenging to control. In the process of smelting and forging, the steel ingot was frequently put into and taken out of the furnace, which would cause the atmosphere to be challenging to control.

### 3.1.2. Cast Iron Covered on Wrought Iron

The second process consisted in covering the cast iron by wrought iron. This process was mainly designed based on Heavenly Creations [18], a technical book written in AD 1637 in the Chinese Ming Dynasty, and some improvements were made in practical operation. In this process, there were a few grooves on the surface of the wrought iron. Several pieces of cast iron were put above the wrought iron. Through this process, the length of the sword was 28.9 cm, and the weight was 192.4 g. The technological process, size, and pattern on the surface are displayed in Figure 2.

Compared with the first process, the second process showed a visible improvement in operational difficulty and efficiency.

1. It improved the contact conditions. The processing of wrought iron ensured sufficient contact between raw materials and provided a better diffusion condition;

2. Preventing the loss of cast iron. Under the condition of high temperature in the furnace, cast iron would fuse rapidly. At this time, fused cast iron would enter into the inside of wrought iron under the influence of gravity, which could prevent the flow loss of cast iron.

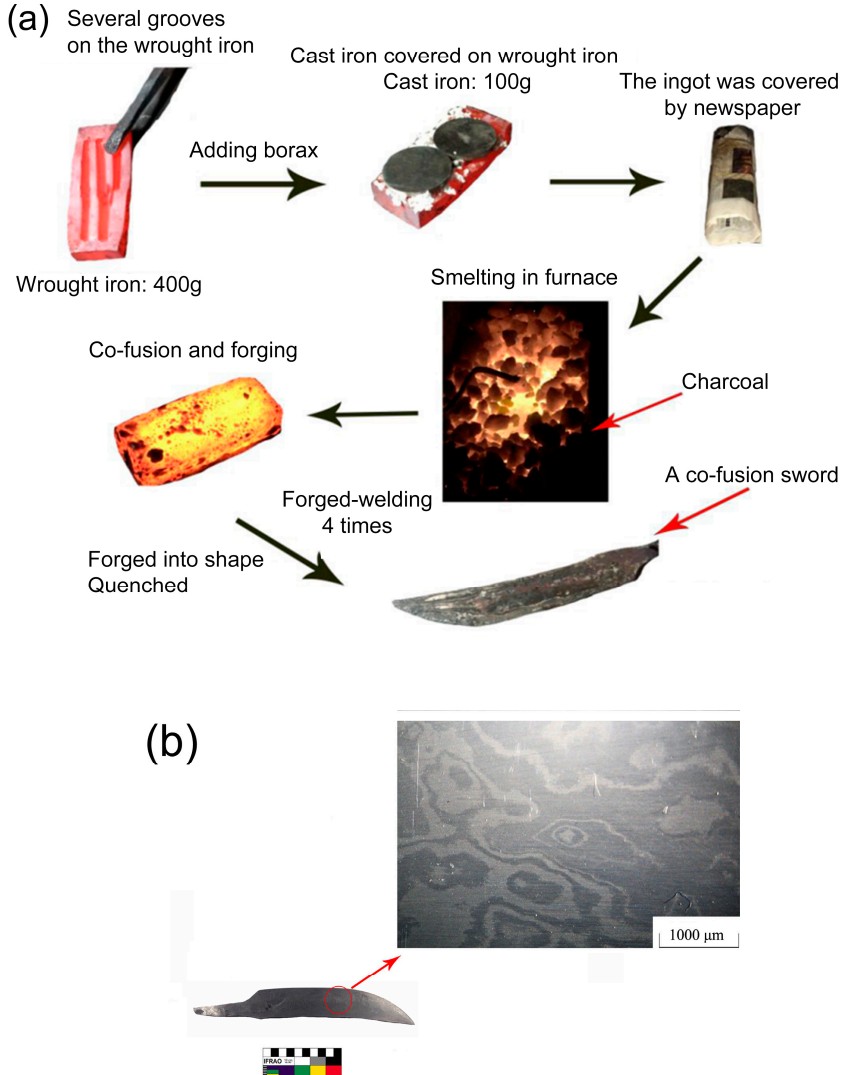

**Figure 2.** Cast iron covered on wrought iron to make swords: (**a**) technological process; (**b**) the sword and pattern on the surface.

### 3.1.3. Inserting Cast Iron into Wrought Iron

The third process consisted of inserting the cast iron into wrought iron. This process was mainly designed based on The Dream Pool Essays [19], a technical book written from AD 1086 to AD 1093 in the Chinese Song Dynasty, and it was improved in practice. In this record, "bending wrought iron, inserting cast iron" meant that the wrought iron was processed into a suitable shape, and the cast iron was inserted into the wrought iron to ensure the formation of a diffusion interface between cast iron and wrought iron. Through this process, the length of the sword was 31.4 cm, and the weight was 221.7 g. The technological process, size, and pattern on the surface were displayed in Figure 3.

This process could be divided into two aspects. From the disadvantages, the process was time-consuming and the clay's internal condition cannot be judged which requires a higher technical level of blacksmiths. The advantages were as follows:

1. The operation was simple, and the internal temperature of the sealed clay was relatively uniform;

2. A relatively independent atmosphere was formed inside the sealed clay. Moreover, the matrix of steel ingot was relatively pure.

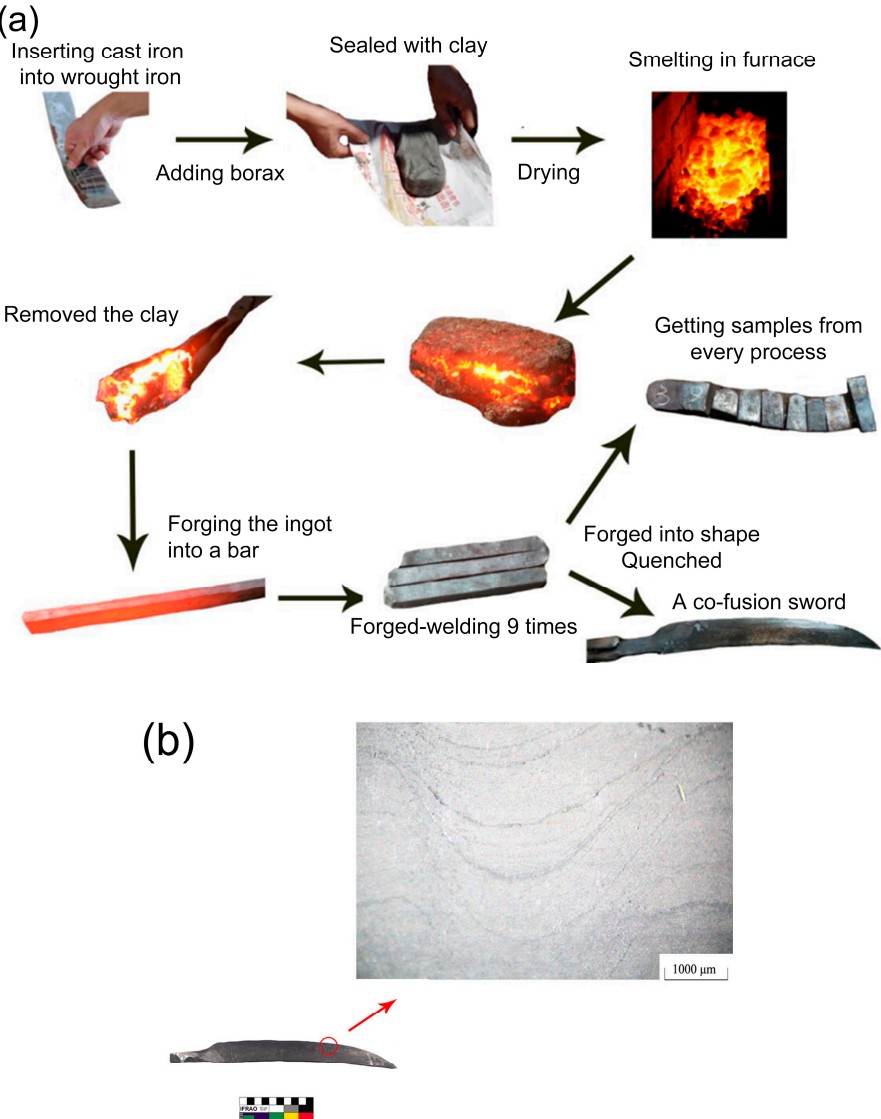

**Figure 3.** Inserting cast iron into wrought iron to make swords: (**a**) technological process; (**b**) the sword and pattern on the surface.

### 3.2. Samples Analysis

#### 3.2.1. Carbon Content and Surface Hardness

Several replicated swords (Groups 1, 2, 3) were sampled and analyzed to study the carbon content and surface hardness. The results were as follows in Figure 4:

According to the results of the analysis, the carbon content and hardness of the three swords were quite different. The carbon content and hardness of Group 1 (co-fusion directly) was significantly lower than the other two groups. It was closely related to the flow loss of cast iron in the co-fusion directly process. Group 3 (inserting cast iron into wrought iron) was slightly lower than that of Group 2, which might be related to forge welding more times.

The carbon content and hardness were analyzed in each process of Group 3. The results are shown in Table 2.

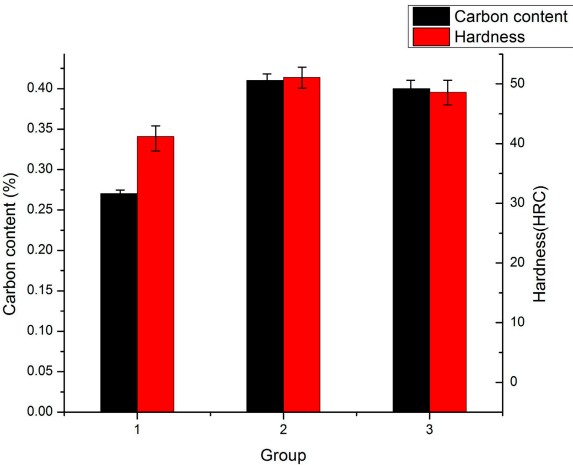

**Figure 4.** Carbon content and hardness of the three swords.

**Table 2.** The carbon content and hardness variety in each process of Group 3.

| Process | | Smelting1 | 2 | 3 | 4 | 5 | 6 | 7 | 8 | Quenching |
|---|---|---|---|---|---|---|---|---|---|---|
| Carbon content wt.% | | 0.59 | 0.56 | 0.52 | 0.51 | 0.46 | 0.47 | 0.45 | 0.43 | 0.40 | 0.40 |
| Hardness/HRC | High carbon layer | 51.6 | 46.8 | 44.2 | 42.4 | 39.2 | 31.8 | 25.6 | 27.2 | 26.5 | 48.6 |
| | Low carbon layer | - | - | - | 2.4 | 11.8 | | | | | |

Through Table 2, it was confirmed that the carbon content had decreased in each process, and an adverse effect on the performance of the co-fusion sword might exist. Therefore, the blacksmith would naturally think of supplementing cast iron in case of excessive forge-welding.

### 3.2.2. Microstructural Observation

To observe the continuous change of microstructural characteristics, we analyzed the samples taken from each process of Group 3. The results of smelting and then forged welding 1 to 8 times are shown in Figures 5 and 6.

Figure 5 was the result of observing the sample at a lower magnification (50×), it showed the continuous evolution of the microstructure on the surface. When forge welding several times, the formation of the pattern was mainly due to the difference in microstructures between the laminated structures. Because the hypoeutectic white cast iron could not be forged directly, the process was to reduce the carbon content in cast iron to the range of malleability then forging repeatedly.

When increasing the magnification of the samples to 200 times (as shown in Figure 6), through continuous observation of the microstructures, the co-fusion samples were transformed from laminated to homogenized. The thickness of the high carbon layers and low carbon layers decreased, then disappeared. It was confirmed that the microstructure in the cast iron transformed to the cementite (Cm) + pearlite (P), the microstructure in the wrought iron transformed to ferrite ($\alpha$) + P. With repeated forge welding, finally, the high carbon layers were mainly composed of $\alpha$ + P.

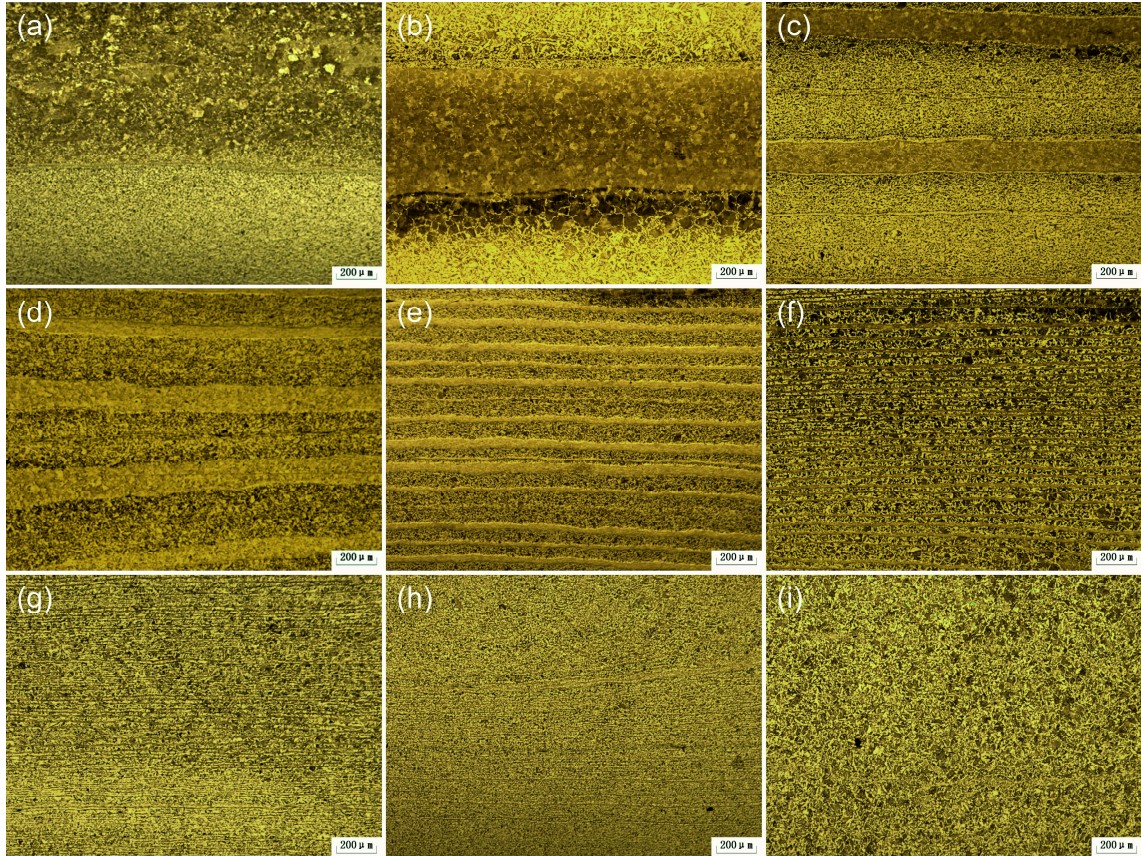

**Figure 5.** Continuous microstructural evolution of group 3, 50 magnification: (**a**) smelting; (**b**–**i**) forge welding 1–8 times.

Carbon diffusion is the most important mechanism for the continuous evolution of the microstructures. Obvious diffusion phenomena could be seen in different layers. In the contact interface between high carbon layers and low carbon layers, the microstructure was displayed that Cm + P, P, P + $\alpha$, $\alpha$ moved from high carbon region to low carbon region. There were much spherical or zonal silicate inclusions on the samples. When forging fewer times, the silicate inclusions mainly reflected as strips, and some granules distributed along the direction of forging. After forging more times, the granule silicate inclusions were mainly distributed in the matrix.

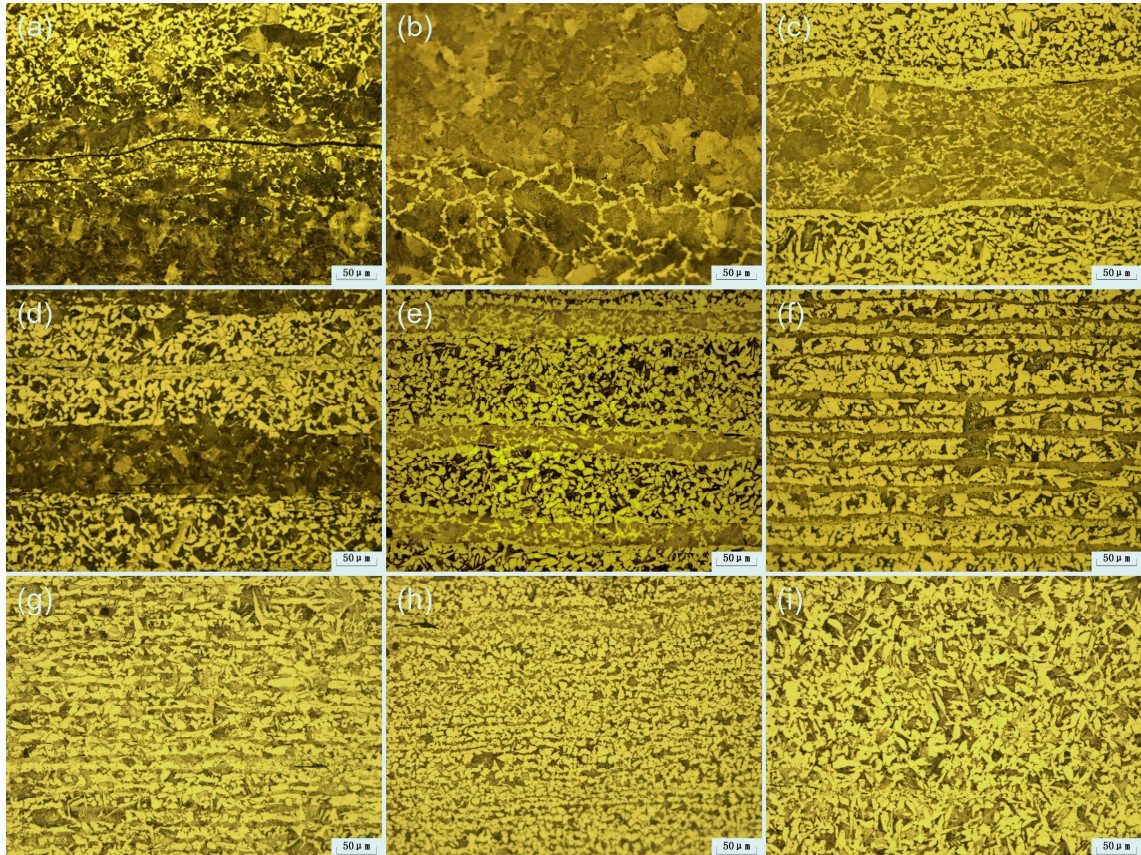

**Figure 6.** Continuous microstructural evolution of group 3, 200 magnification: (**a**) smelting; (**b–i**) forge welding 1–8 times.

## 4. Discussion

### 4.1. Microstructures of Simulation Co-Fusion Samples in Different Procedures

There were apparent differences in several procedures due to the continuous analysis of the microstructures of the co-fusion samples. They could be divided into three cases: the initial state, forge welding for several times, and homogenized.

The initial state meant that the interface between cast iron and wrought iron was formed, and the carbon diffusion was still in progress. Therefore, the lamination of materials and apparent carbon diffusion were the main microstructure characteristics of the samples in the initial state. Figure 7 was the result of materials co-fused for several minutes, so the information of the internal cast iron and diffusion layer was well preserved.

In Figure 7, it can be seen that there was an apparent laminated phenomenon in the co-fusion sample. The microstructure can be divided into four layers from the inside to the outside:

1.　Primary cast iron layer: the microstructure of this layer was mainly pearlite and network carbide which was the same as hypoeutectic white iron;
2.　High carbon transition layer: due to the carbon diffusion in cast iron, the microstructure of this layer mainly consisted of pearlite and acicular carbide;
3.　Low carbon transition layer: the carbon content of this layer was further reduced, and the microstructure was pearlite and ferrite;
4.　Primary wrought iron layer: the microstructure of this layer was mainly ferrite.

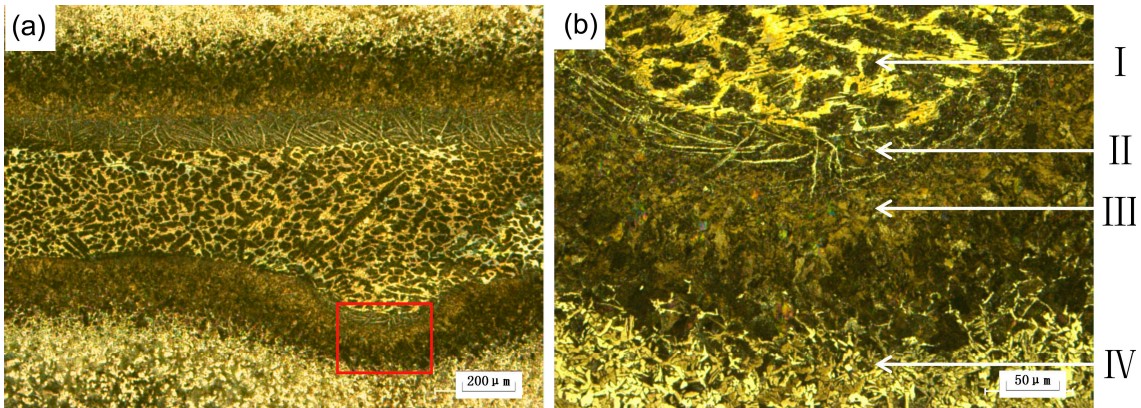

**Figure 7.** Microstructure of the initial state in co-fusion: (**a**) 50 magnification; (**b**) 200 magnification.

After forge welding for several times, the co-fusion samples would show a state of uniform lamination. Compared with the initial state, this lamination was mainly composed of high carbon layers and low carbon layers, as shown in Figures 5 and 6, and there was no residual cast iron microstructure observed. The microstructure in the layers was different after forging, which was directly related to the proportion of materials before smelting.

The distribution of inclusions was quite different between the layers, as shown in Figure 8. The matrix in primary cast iron layers was pure, and the inclusions were mainly distributed at the boundary. This phenomenon was not observed in primary wrought iron layers.

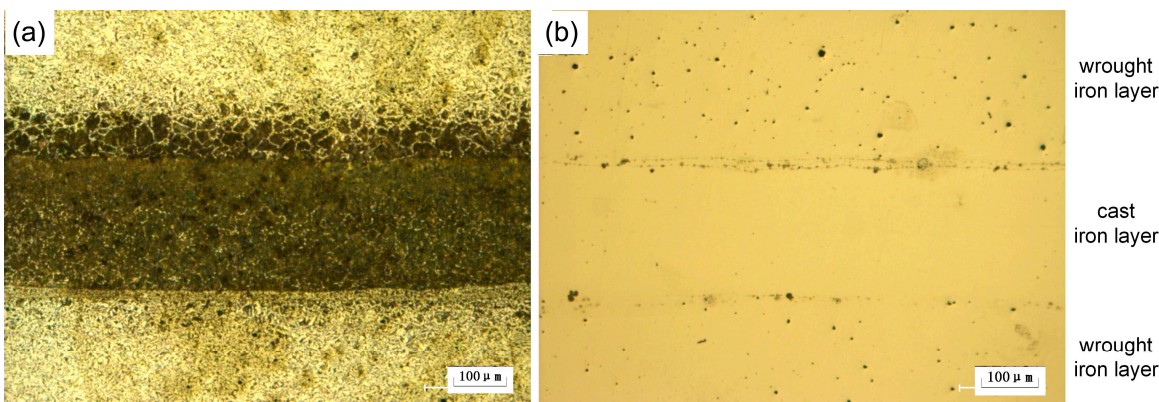

**Figure 8.** Distribution of inclusions in laminated co-fusion samples: (**a**) etching with 3 vol.% nitric acid for 5 s; (**b**) no etching.

Homogenized co-fusion steel could be obtained after forge welding numerous times. It was difficult to see the laminated microstructure because of the massive carbon diffusion between the layers. The inclusions were also evenly distributed. When the number of forging layers is tens of thousands or even hundreds of thousands, the carbides are uniformly distributed, and individual layer discretion is lost. Therefore, the homogenized co-fusion artifacts could hardly reflect the microstructure characteristics mentioned before.

### 4.2. Archaeological Artifacts Identified as Being Made by Co-Fusion Steelmaking

Combined with the micro-analysis of previous studies [14–17,20], the samples identified as co-fusion artifacts were concentrated in the initial state. Several archaeological artifacts identified as made by co-fusion steelmaking could be divided into two categories:

The first category was the final products such as swords and tools. Iron artifacts in References [14,15,17] fell into this category. An evident laminated phenomenon in these samples could

be observed, and the carbon diffusion between the layers existed. These artifacts were forged into shape, but the quality was only on an average level. Therefore, they could not represent high-quality co-fusion products.

The second category was the intermediate artifacts, and the shape of them was irregular. Iron artifacts in References [16,20] had some shrinkage cavities and other defects. These characteristics indicated that the iron artifacts were likely to be the defective products that failed in forging. Therefore, they could better preserve the information of residual cast iron.

Interestingly, there were no homogenized iron artifacts identified as made by co-fusion steelmaking at present. This phenomenon hardly corresponded with the abundant records of co-fusion in ancient Chinese literature. The key to this problem lays in the lack of understanding of the microstructure of homogenized co-fusion artifacts.

Therefore, it showed that the current understanding of co-fusion was far from enough, and several co-fusion iron artifacts were identified as manufactured by other processes. Observation of the microstructure of the simulation samples provided a basis for rethinking about this problem. It is crucial to understand the development of ancient steelmaking technology.

### 4.3. Ancient Chinese Co-Fusion Artifacts Transformed from Laminated to Homogenized

The change of microstructures in the forge welding repeatedly provided a basis for further understanding of ancient co-fusion samples. The most crucial point was the transformation from the laminated structure to the homogenized structure. For this process, when forging a few times, the number of real layers could be the same as the layers observed by the naked eye or a metallographic microscope. However, several primary layers disappeared when forging more times, and the materials tended to be homogenized. After forge welding a great many times, homogenized steel with excellent properties could be obtained.

Laminated steel had a fascinating history. In ancient China, the most famous laminated steel was "bailian" steel [21,22]. The experimental results showed that, after forge welding several times, the co-fusion samples had similar microstructure characteristics with that of "bailian" steel so there were some problems in identifying the microstructure of ancient artifacts in the past. The laminated structure was often shown as patterned steel on swords which some scholars also called "Welded Damascus" steel [23].

However, after repeated forge welding, there were similar laminated structures. Perhaps, the difference in technology was closely related to the fact that the co-fusion process paid more attention to efficiency. The sword-making techniques of different periods in ancient China were also quite different. In the earlier literature, there were many legends about the combination of co-fusion and "bailian" steel to make swords. While in the later literature, the Japanese swords in the East and the Damascus swords in the West were more respected [24]. This phenomenon reflected the change of ancient Chinese sword-making technology.

Homogenized steel was a high-quality steel material pursued in ancient times. This kind of material could avoid the cracking of steel from the junction of different layers during pressure processing and had excellent mechanical properties for making weapons and tools. This experiment showed that co-fusion could make steel with different carbon content into homogeneous materials, which provided an essential basis for ancient China.

Forge welding quantity times after co-fusion could cause relatively massive carbon loss. Ancient blacksmiths might find this phenomenon and add crushed cast iron pieces in every co-fusion and forging process to ensure their swords' mechanical properties. This method conformed to the ancient Chinese literature repeated co-fusion, just as "co-fused thousands of times" [1].

## 5. Conclusions

We replicated co-fusion swords in a sword workshop by different processes and analyzed the microstructural changes of the samples during the repeated forge welding process. The following conclusions were obtained:

1. Based on ancient Chinese literature, several processes could be used to replicate the co-fusion swords with excellent performance. However, there might be different processes, and the simulation experiments could not completely represent the co-fusion steelmaking;

2. After repeatedly forge welding and homogenization annealing, the steel ingot with a uniform carbon content could be obtained. In co-fusion steelmaking, the microstructures transformed from composited to homogenized because of the forge welding process. These results prompted people to consider the micro-characteristics of archaeological co-fusion artifacts. It was of great importance to the understanding of ancient steelmaking technology.

**Author Contributions:** Conceptualization, S.Q. and W.Q.; methodology, S.Q. and W.Q.; software, S.Q.; investigation, S.Q.; writing—original draft preparation, S.Q.; writing—review and editing, W.Q. All authors have read and agreed to the published version of the manuscript.

**Funding:** This research received no external funding.

**Conflicts of Interest:** The authors declare no conflict of interest.

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
