# Peer review of "Replication Experiments and Microstructural Evolution of the Ancient Co-Fusion Steelmaking Process"

_metals, doi:10.3390/met10091261_

Round 1

Reviewer 1 Report

The English quality is such that reading the paper is difficult. See highlighted words and expressions and sometimes suggestions in the text. Watch for expressions like forging welded, which should be forged-welding

Why do you call the processes at the core of the paper co-fusion? Is there fusion (melting, probably of pig iron) taking place as the word indicates? From what I read, the answer is no. Please explain this contradiction. 

You seem to distinguish between 4 processes: 3 are the so-called co-fusion variants and the 4th is continuous forge welding. But to me, the 4th process is not different from the first 3.

The mechanisms involved are only hinted at, not desired as such. Carbon diffusion is essential in explaining what is taking place and it would deserve some lip service, maybe even some quantitative description using a mathematical model. 

You also explain that some variants of the co-fusion technique require more skills from the blacksmith: can you explain why and go more into details?

The various processes are described only qualitatively: what temperatures are obtained for example the furnace? What is the atmosphere? The point of simulating ancient processes is to be able to monitor them with appropriate measuring devices. 

At the end of the day, did you analyze the origin of each archaeological sample in terms of which process? I did not follow you argument very clearly.

Last, what is the difference between what you describe and Dasmacus steel in the Mediterranean or the Katana in Japan? There are similar stories in Europe for example. 

Reviewer 2 Report

The methodology should be expanded and hardness and carbon measurements described in detail.

How many hardness measurements were made? How was the carbon content measured? Are the values of Carbon content ​​given in atomic percentages or by weight?

For a full analysis, it would be necessary to perform, for example, measurements of fracture toughness, plastic properties – these properties are important in such applications.

The font in fig. 1-3 must be enlarged - in this form the drawings are illegible

Table 2 should be corrected

In fig. 4, the error bars (ex. Standard deviation) should be marked

In my opinion, the discussion should be expanded further, but with such a limited scope of research, it will be difficult.

The latest literature item is dated 2015, and the list contains only 18 items - it would be good to epand.

Reviewer 3 Report

Original and interesting subject of the work, devoted to ancient China methods
process of
swords manufacturing, but research of the work leaves a deep
dissatisfaction. Microstructure and properties analysis is limited only
to the use of light microscopy and Rockwell hardness tests
(the parameters of the Rockwell hardness tests are not given), thus
the work should be qualified as initial research.
Although the light microscopy observations
are interesting, are only preliminary results which have to be extended to
provide
a full description of the received
materials, that's why I cannot recommend
the manuscpirtit for publication.
To be published in scientific journal, as Materials,
the microstructural research has to be extended by (at least): the chemical composition
analysis (micro and macrosegregation of elements), phase composition analysis
(XRD), SEM microscopy obsevations accompanied by EDS analysis (maps), mechanical properties
examination ...

Reviewer 4 Report

In my opinion the article is interesting study of co-fusion steelmaking based on Chinese literature. The replication of co-fusion swords has been performed and the carbon content and surface hardness were investigated. It allow to find differences between the used methods. The historical informations  will be brought closer to readers.

Author Response

First, we want to thank reviewer 4 for the careful read and accurate comments on previous draft.

Point 1: The replication of co-fusion swords has been performed and the carbon content and surface hardness were investigated. It allow to find differences between the used methods.

Response 1: Reviewer 4 accurately evaluated the core of this research.  

Point 2: The historical informations will be brought closer to readers.

Response 2: The authors would thank reviewer 4 for the understanding of this research. Different from the research of modern metals,  the research on ancient iron and steel materials mainly lies in excavating its historical value. And it is essential to solving the issues in the history of steelmaking technology.

Round 2

Reviewer 1 Report

Lines 26-28: "The metallurgical system based on cast iron laid a continuity of civilization and a solid foundation for the national integration and unity, and the continuity of civilization in ancient China[1.]." I am sorry, but this statement is "a priori" and not based on any rationale that you have presented.  I disagree with you: the development of China is likely to have been driven by the availability of cast iron, i.e. liquid hot metal in modern English. True, it is a major difference with the development of steel technology in the West, but what is the historical significance of this difference?? I'd suggest that you read Bruno Latour and his work. 

I am not sure that reference 10.1007/s11837-014-0880-8 is relevant to your paper. 

You added a set of papers related to ancient steel in the West (Mediterranean) and referred to them quickly in the initial paragraphs.  One paper at least (above) is irrelevant, as Iron Age iron has not much to do with Damascus material!!!

The most relevant book on Damascus steel, as far as I know, is "Damascus Steel, Myth, History, Technology, Applications, Manfred Sachse, 2008, Stahl u. Eisen Verlag". You might want to compare it with your work and mention it in your text?

However, what is exactly your point in discussing similarities and differences between co-fusion and Damascus? 

It seems to me that on the one hand, elsewhere (the West, India, Japan), the technology of making steel was entirely based on solid production. On the other hand, China was unique in as far as liquid hot metal (cast iron) was available. This may be the major reason why elsewhere Damascus steel blades were invented, while in China the development went further and it was possible to produce a homogenized structure : thus quite different materials (composite vs. homogeneous material) for the same purpose of making blades. 

289-295: what is your point??? Not clear! What were these ancient sword making techniques compared to forged-welding? What does it mean that western and japanese technologies were regarded with more respect???

Lines 307-309: "Replicate the co-fusion swords in a sword workshop by different processes. Analyzing the microstructure changes of the samples during the repeated forging welded process." These sentences do not have a functioning verb!!! I imagine you want to summarise the content of your paper? "We replicated the co-fusion .../..."

What does this mean??? (310-312) "Based on ancient Chinese literature, several processes could be used to replicate the co-fusion swords with excellent performance. However, there might be plenty of different processes, and the  simulation experiments could be included in them."

The next paragraph is also incomprehensible! Should be carefully rewritten!!! I had suggested that you asked a native speaker. 

You also did not change "forging-welded" into "forged-welding" and "forged-welded" respectively in lines 313 and 315. But there are more occurrences left in the rest of the paper. You changed the initial expressions but left the ones that I did not indicate to you (underlined) unchanged. 

At the end of reading your revised paper, I am not clear as to what you are trying to say! Your experiments are to a large extent fairly clear but the discussion regarding what is called forge-welding in the "literature" and the simulations you made of it in your experiment is confusing. And the discussion of Damascus material is also confusing. 

Reviewer 2 Report

I would like thank you for your response.

I think that you shoud improve the manuscript. You didn’t enlarged the fonts in figs: 1, 2, 3, 5, 6, 8, 9, they are too small – figs are no readable. You only enlarged the size of the pictures. Additionally, there is no scale bar in fig. 7.

As I wrote last time, the scope of research should be extended (e. g mechanical investigations, deeper analysis of the microstructure with the higher magnification or other techniques), then the discussion can be expanded, and the article will have scientific soundness.

Reviewer 3 Report

The work is still under early research, and the title change does not
explain the very limited experimental research done on the manufactured
replicas. I would like to point out that the Reviewer did not require
and did not recommend microstructural tests of original swords (museum
objects), as the authors suggest in the coverletter.

Despite the linguistic corrections and supplementing the description of
the experiments,the work has still many shortcomings and is not ready
for publication. In the corrected version of the manuscript, the characteristics
of obtained materials (replicas of swords) are still incomplete, so again
I recommend completing the research in this area and I do not recommend
the work for publication.
As the characterization of the produced
materials (replicas) has not been made properly, therefore the analysis
of the influence of the production method (and its parameters) on the
microstructure and properties of the replicas cannot be properly carried
out. The main reason for this is the imperfection of the experimental
procedure
in the microstructure examinatoion. The planned and conducted
experimental tests are insufficient to
ensure a basic level of scientific
quality of the work.
Only after completing the characteristics of the
materials(replicas), which Authors declare in the Coverletter, the work
can be resubmitted.

Round 3

Reviewer 2 Report

I would like thank you for your response. I accept the article in the present form.

Reviewer 3 Report

The subject of the work is interesting, a lot of effort was put into recreating the process of manufacturing of swords with the microstructure and properties of ancient China swords. The text of the work has been redrafted and re-organized, which significantly improved its readability. Most of the linguistic errors have been corrected, although there are still some single editing errors.
Unfortunately, the microstructural studies of the manufactured replicas were not extended, which was already suggested in the first review (point 4). Microstructural analysis is still carried out using only light microscopy, the basic method in materials research, which has its limitations. The study lacks full identification of the phase composition (quantitative analysis) as well as the analysis of changes in the chemical composition between layers and phases visible in the analyzed areas. Without which, the information about the microstructure of obtained materials is incomplete, and therefore drawing any conclusions as to the influence of the process of manufacturing on the microstructure of the replica's materials is pointless. Therefore, I do not recommend the work for publication again and again, I recommend supplementing the microstructural studies with X-ray phase analysis (XRD), scanning electron microscopy (SEM observations are possible at relatively low magnifications of 100x, 200x ...), along with the EDS analysis (elemental distribution maps). These research methods are simple and fast, and I'm sure they provide valuable information on the microstructure changes in replica materials received, which will significantly increase the scientific value of the work (necessary for publication in the Materials ). EDS analysis results allow to discuss the diffusion of carbon between layers or between phases or microstructural components, which the Authors are currently undertaking only on the basis of images taken by a light microscope (which in my opinion is insufficient).